# Application of Double Piping Theory to Parallel-Arrayed Low-Pressure Membrane Module Header Pipe and Experimental Verification of Flow Distribution Evenness

**DOI:** 10.3390/membranes12070720

**Published:** 2022-07-20

**Authors:** No-Suk Park, Sukmin Yoon, Woochang Jeong, Yong-Wook Jeong

**Affiliations:** 1Department of Civil Engineering, Engineering Research Institute, Gyeongsang National University, 501, Jinju-Daero, Jinju 52828, Korea; nspark@gnu.ac.kr (N.-S.P.); gnuysm@gmail.com (S.Y.); 2Department of Civil Engineering, Kyungnam University, 7, Kyungnamdaehak-ro, Masanhappo-gu, Changwon 51767, Korea; jeongwc@kyungnam.ac.kr; 3Department of Architecture, Sejong University, 209, Neungdong-ro, Gwangjin-gu, Seoul 05006, Korea

**Keywords:** double piping theory, flow distribution evenness, manifold pipe, membrane module, computational fluid dynamics, ultrasonic flowmeter

## Abstract

In this study, the improvement effect of flow distribution evenness is evaluated quantitatively by applying the double piping theory to a parallel-arrayed low-pressure membrane module header pipe structure, and its feasibility is discussed. Orifice inner pipes to be inserted into a full-scale membrane module header pipe are designed via the computational fluid dynamics (CFD) technique, and the flow rates into 10 membrane modules are measured in real time using a portable ultrasonic flowmeter during operation to verify the effect. Results of CFD simulation and actual measurements show that the outflow rate from the branch pipe located at the end of the existing header pipe is three times higher than the flow rate from the branch pipe near the inlet. By inserting two inner pipes (with an open end and a closed end into the existing header pipe) and applying the double pipe theory, the flow distribution evenness is improved. The CFD simulation and experimental results show that the flow uniformity can be improved by more than 70% and 50%, respectively.

## 1. Introduction

Parallel-arrayed low-pressure membrane processes, such as microfiltration and ultrafiltration, have been increasingly applied to drinking water treatment systems. These low-pressure membrane processes selectively remove particulate contaminants larger than the pores through a straining mechanism via the differential pressure between the membrane pores [1,2,3]. Membrane filtration is used extensively as it can remove particulate matter, organic matter, inorganic salts, etc., as well as yield stable water quality, depending on the pore size [4,5]. However, despite the many advantages of membrane filtration technology, the decrease in filtration efficiency over the operating time (i.e., the periodic occurrence of membrane fouling) has been identified as a significant problem in the introduction of membrane filtration technology to water treatment fields [4,5,6,7]. Generally, raw water flows into each membrane module installed parallel to the header pipe. The filtered water treated by the membrane accumulates at the top and flows out (see Figure 1). Its structure enables the treated water to flow from a relatively large header pipe at the bottom to a membrane module installed in parallel through upward manifold pipes [1,2,3].

This structure has been widely applied to thermodynamic cooling. In the early 1990s, Shen et al. and Datta and Majumdar [8,9,10] introduced a method for introducing refrigerants uniformly into cooling systems. They experimentally deduced that a smaller ratio of the cross-sectional areas of the manifold branch pipes to that of the header pipe results in a more even flow rate distribution. In addition, Eguchi et al. [11] discovered that a smaller ratio of the cross-sectional area of the manifold to that of the header pipe results in a smaller loss factor (the ratio of the average power loss to the peak load loss).

However, this method cannot be applied easily to the membrane processes for water treatment. Because the inlet diameter (diameter of the manifold branch pipe) of commercial low-pressure membrane modules is typically 50 mm or greater, the reduction in the ratio of the header pipe to the cross-sectional area of the distribution manifold to be connected is limited. When the ratio is small, the head loss and energy consumption increase. According to numerical analysis, Hong and Riggs [12] emphasized that a tapered header pipe with a gradually decreasing cross-sectional area achieves a more uniform flow velocity distribution than a distribution pipe with a constant cross-section; hence, the hydraulic pressure becomes constant. However, this method is applicable only when the flow velocity of the header pipe is relatively low. Meanwhile, header piping applied to an actual membrane filtration process for drinking water treatment is generally designed and operated to maintain a flow rate of 1.0 m/s or more. Muhana and Novog [13] investigated the effect of the header pipe flow rate and Reynolds number on the flow distribution through each manifold branch pipe for a header pipe. Their experimental results showed that, as the Reynolds number increased, the flow of the manifold branch pipe on the far side from the header pipe inlet increased. In addition, considering that a header pipe exhibits the characteristics of pipe flows, Park et al. [1] used the computational fluid dynamics (CFD) technique to determine the relationship between the Reynolds number and the evenness of the manifold flow distribution in a header pipe. In cases involving Reynolds numbers of 4000 or less, the flow into each parallel membrane module from branch pipes was relatively even.

Since the late 2000s, studies have been conducted to optimize the designs in the fields of water treatment and supply, where the manifold is applied via CFD simulations [14,15]. By performing CFD simulations and verification experiments on modular membrane systems, Ding [16] investigated the occurrence of significant unevenness in the flow rate from a header pipe to each membrane module. In addition, On the basis of CFD simulation results, Paul et al. [17] inferred that the flow rate of a fluid flowing into parallel-arrayed unit cells in a stack of proton exchange membrane fuel cells is associated with the flow direction of the outflow header. According to CFD simulations and pilot-scale multipipe experiments, Kim et al. [2] proved that a double piping structure with an orifice pipe inserted inside a header pipe can evenly distribute the flow rate.

Therefore, in this study, double-header pipe structures were designed for the water treatment membrane filtration process. The designed structures were optimized via CFD simulations. According to two optimized design concepts, orifice inner pipes were manufactured and installed in an inflow header pipe for insertion into an existing header pipe. The flow rates into the parallel-arrayed low-pressure membrane modules were measured, and the effect was experimentally verified. Accordingly, the feasibility and efficiency of the dual piping theory for a parallel-arrayed low-pressure membrane module header pipe were evaluated quantitatively.

## 2. Research Methods

### 2.1. Membrane Filtration Process for Drinking Water Treatment

The membrane filtration facility selected for this study was located within the G treatment plant in the Republic of Korea, with a maximum capacity of 30,000 m^3^/day. The specifications of the membrane are listed in Table 1. The membrane was a microfiltration membrane. The membrane filtration flow rate (Flux) was designed to be 1.0 m^3^/m^2^·day under normal conditions, and, when one series was terminated for backwashing, the filtration flow rate of the remaining three series was increased to 1.33 times during normal operation to maintain a constant flow rate. The membrane filtration system was equipped with four series, 24 units (six units per series), and 480 modules (20 membrane modules per unit), where each unit comprised 20 modules arranged symmetrically (see Figure 1 and Figure 2). The membrane filtration process was operated with 50 s of water flow from the inlet, 30 min of membrane filtration, 30 s of backwashing (air + water), 45 s of drainage, and chemical enhanced backwashing; furthermore, clean-in-place instructions were conducted periodically.

Figure 2 shows the lower section of the inlet header pipe. Ten membrane modules were installed at an interval of 0.40 m at a position 0.35 m from the inlet; the distance between the last tenth of the pipe and the end of the header pipe was 0.35 m, as in the inlet section. The average daily operating flow for the inlet header pipe unit was approximately 35.96 m^3^/h.

### 2.2. Methodology of CFD Simulations

In this study, ANSYS CFX16.0, a commercial CFD code, was used to simulate the fluid behavior of an existing header manifold pipe (Figure 3a), a double-header manifold pipe with an orifice inner-end open pipe (Figure 3b), and a double-header manifold pipe with an orifice inner-end closed pipe (Figure 3c). The CFD simulation was performed to predict the extent to which the flow rate into each membrane module was evenly distributed as compared with the case for an existing header manifold pipe before installation, fabrication, and verification. The authors investigated the adequacy of the proposed double-header manifold pipe designs (Cases A and B). Cases A and B were simulated by changing the diameter of the orifice installed in the inner pipe through trial and error. As shown in Figure 3b,c, the diameter of orifices No. 1 to No. 7 was set to 65 mm, whereas 60 mm was set for orifice No. 8, 40 mm was set for orifice No. 9, and 20 mm was set for orifice No. 10.

Park et al. [18] introduced double piping theory to a pilot-scale manifold pipe for improving the evenness of the flowrate to each branch pipe by inserting an inner orifice pipe into the header pipe. They revealed that inserting an inner orifice pipe within the original single header pipe caused the inlet water to turn at the end of the right side. The friction caused the head loss at the left side to be higher than at the right side as the inlet water flowed from left to right. The head gradient reversed after the main flow turned at the end of the right side, and then flowed right to left. The head loss and energy level at each branch pipe could be kept constant because the water passed through the orifices in the inner pipe.

The length of the pipe was 4.3 m in all three cases, and the distance between the branch pipes and inner pipe orifices was the same for all. The inner tube diameter of Cases A and B was 83.1 mm. The branch pipe outlet was directed horizontally toward the left and right. A steady state was assumed in the CFD simulation. In terms of the characteristics of the fluid behavior in the flow field, an inflow water temperature of 25 °C was assumed (i.e., based on the room temperature).

The CFD simulation was performed by segmenting the geometry of interest into numerous elements, which are collectively known as grids or cells. Subsequently, momentum and continuity equations were formulated for each grid according to the specified boundary conditions, and they were repeatedly solved using the finite volume method.

The time-averaged Navier–Stokes equations for momentum and continuity were solved to achieve a steady, incompressible, turbulent, and isothermal flow. The continuity and momentum equations are expressed as follows:(1)$$\underline{\nabla }\cdot \left(\underline{U}\right)=0,$$
(2)$$\underline{\nabla }\cdot \left(\rho \underline{U}⊗\underline{U}-\mu \underline{\nabla U}\right)=\underline{B}+\underline{\nabla }P-\underline{\nabla }\cdot (\bar{\rho \underline{u}⊗\underline{u}}),$$
where ρ and μ are the fluid density and dynamic viscosity, respectively, *P* is the pressure, *U* is the fluid mean velocity, *B* is the body force, and *u* is the fluctuating velocity.

The authors assumed that turbulence in the pipes was isotropic. Therefore, a standard k–ε model was used to model the turbulence transport of the momentum. A no-slip condition was assumed at the pipe wall surface, and the widely used standard wall boundary method was applied to bridge the viscous sublayer. Therefore, the velocity of the component at each wall was assumed to be zero. The wall shear stress was obtained based on the logarithmic law of the wall [1].

### 2.3. Flow Distribution Measurement Using Ultrasonic Flowmeter

Two inner orifice pipes were manufactured (see Figure 4a) and installed in an existing header pipe. While operating the target membrane unit normally, the flow rate into the 20 membrane modules was measured, and the effect was experimentally verified (see Figure 4b). In this study, the flow rate into each module was measured using a clamp-on type (dry-type method) ultrasonic flowmeter. An ultrasonic flowmeter measures the flow rate in a pipeline according to the characteristics of ultrasonic waves. Recently, the application field of the clamp-on type (dry-type method) ultrasonic flowmeter has been expanded owing to its easier installation compared with the wet-type method, as well as its easy management owing to no damages to the pipe. The wet-type method affords a relatively better accuracy than the dry-type method. However, the dry-type method is preferred because of its fewer installation and mobility issues. Table 2 summarizes the specifications of the transducer and the flowmeter used in this study.

To measure the flow rate accurately, a distance of 5D or more from a curved pipe is required [16]. However, the distance between the branch pipes was only approximately 0.40 m for the selected membrane module inlet header pipe. Therefore, accurate flow rate data were difficult to obtain. As illustrated in Figure 3b, the flow rate data were read directly from the header pipe, and the amount of change in flow that occurred as it passed through each branch pipe was regarded as the inflow flow rate of the branch pipe. As shown by the measurement results, the flow rate data were relatively unstable; however, stable data were obtained approximately 5 min after the installation.

## 3. Results and Discussion

### 3.1. Results of CFD Simulations

Figure 5 presents the simulation results obtained via CFD simulations when the inflow flow rate was 35.96 m^3^/h for the inlet manifold pipe structure (Figure 3a) used in the membrane filtration process. The inlet was located the lower right section, and the main flow of raw water was distributed while it was flowing to the upper left section. Figure 5a shows the velocity field as a vector, and Figure 5b shows the pressure as a contour. As shown in the figure, the area closer to the inlet (lower right) indicated a relatively high velocity exceeding 1.0 m/s. However, toward the end of the header pipe (upper left), the velocity decreased to less than 0.1 m/s. As shown by the pressure distribution in Figure 5b, the pressure increased toward the end of the upper left header pipe. Assuming that the permeability coefficients (K) representing the resistance of each membrane module are almost identical, the phenomenon above was attributable to Darcy’s law. Considering the pressure difference (head loss) between the inlet and outlet of the membrane module, the head loss was approximately twice as large in the branch pipe located at the end of the header pipe compared with the branch pipe at the inlet side of the header pipe. On the basis of Darcy’s law, the velocity (V) through the membrane module increased; hence, the flow rate increased toward the end of the header pipe, according to the continuity equation shown in Equation (3).
V = Q/A = −K dh/dL,(3)
where V and Q are the flow velocity and flow rate, respectively, A is the cross-sectional area of the membrane module, K is the permeability coefficient of the membrane module, dh is the pressure difference (differential pressure) between the inlet and outlet of each membrane module, and dL is the length of the membrane module.

The pressure difference generated between the membrane module toward the end of the header pipe contributes to the increase in the branch pipe flow rate toward the end of the header pipe instead of near the inlet in the parallel-arrayed membrane modules. The method of achieving an even flow distribution by applying a tapered header pipe with a reduced cross-sectional area, which has been proposed in the energy engineering field [12], cannot be applied to an actual water treatment membrane module piping structure. In the case of a tapered header pipe, it can be applied when the flow rate in the pipe is constant with respect to time. It is not applicable when the flow rate varies with time due to membrane fouling. In addition, if the flow velocity in the head pipe is turbulent, a tapered header pipe with reduced hydraulic cross-sectional area cannot be applied.

Figure 6 and Figure 7 show the CFD simulation results of a double-header manifold pipe with an orifice inner-end open pipe (Figure 3b) and a double pipe with an orifice inner-end closed pipe (Figure 3c). In both cases, the flow velocity or pressure in the inner pipe changed, but the velocity and pressure of the external header manifold pipe showed a relatively stable flow pattern compared with the existing pipe. To investigate this more comprehensively, the flow rate pattern of the manifold pipe was investigated via CFD simulation, and the results are shown in Figure 8.

Figure 8 shows a comparison of the distribution of flow into the 10 membrane modules through manifold pipes. The flow rate is expressed as the mass flow (kg/s). For the existing pipe, the flow rate from the manifold pipe increased toward the end. This was due to the increase in the pressure difference toward the end of the header pipe, as well as the standard deviation of the outflow rate of each manifold (which was calculated to be 2.126 kg/s). For the double-header pipe (Case A), the outflow rate showed a negative value at outlet No. 1 owing to its high velocity. From the second outlet to the last outlet, a relatively even flow rate was observed, as compared with the case of the existing header piping. For Case A, the standard deviation of the outflow rate was calculated to be 0.6543 kg/s. On the basis of the CFD simulation results, Case B showed the most even distribution of the outflow rate. A high flow rate was observed at the No. 10 outlet, but the overall standard deviation was 0.579 kg/s. The standard deviation obtained from the CFD simulation results alone was insufficient for quantitatively determining whether Case A or Case B was more effective (even) in terms of the outflow distribution. Therefore, we attempted to obtain a more effective design through verification experiments.

### 3.2. Verification Experiment Results

In this study, the inner orifice pipes of Cases A and B were inserted into manifold header pipes located at positions symmetrical to the existing header pipe. Table 3 shows the results of the flow rate measured from the manifold outlet using an ultrasonic flowmeter. The measured absolute values differed slightly from the CFD simulation results, whereas the evenness of the outflow rate was similar between both results. The standard deviations for the existing header pipe, Case A double pipe, and Case B double pipe were 0.54, 0.287, and 0.33 m^3^/h, respectively. According to the CFD simulation results, the standard deviation of the outflow through the manifold was lower in Case B as compared with that in Case A. However, the verification experiments showed that the standard deviation of Case A was lower than that of Case B. In all three cases, a low flow rate was observed at outlet No. 1. This was attributable to the close proximity of the junction (reducer) with the existing pipe to outlet No. 1 during the installation of the reducer, whose diameter was equal to that of the inner pipe; consequently, the flow velocity near the inlet was extremely high. Therefore, the flow rate passing through outlet No. 1 was relatively low. In the existing header pipe, a low outflow rate was observed at outlet No. 1, which was near the inlet, owing to the abovementioned reason (see Figure 9). This scenario could not be improved via the verification experiments on the membrane modules during an actual operation. However, this problem could have been sufficiently prevented if the introduction of such a double pipe was considered in the design stage.

In all three cases, the flow rates of outlet Nos. 7 and 8 decreased abruptly. A similar pattern was observed from the CFD simulation results of Cases A and B, as shown in Figure 8. However, two issues need to be solved regarding this phenomenon. First, in the optimal design of the orifice diameter of the inner pipe, the actual conditions of the pipe may not be reflected. Second, the flow may become unstable, as the point where the inlet wave and the reflection wave of the water flowing through the header pipe coincides at approximately one-third of the pipe length (the locations of outlet Nos. 7 and 8).

## 4. Conclusions

In this study, double-header pipe structures were designed for the water treatment membrane filtration process. The designs of these structures were optimized via CFD simulation. Following two optimized design concepts, orifice inner pipes were manufactured and installed in an inflow header pipe for insertion into an existing header pipe. The flow rates into the parallel-arrayed low-pressure membrane modules were measured, and the effect of improving the evenness of the flow distribution was experimentally verified. The results are summarized as follows:For the parallel-arrayed low-pressure membrane module inflow manifold header piping, the outflow increased toward the end of the header pipe instead of near the inlet. The outflow rate from the first branch pipe in the inlet and that from the branch pipe located at the end of the header pipe differed by three times or more. The increase in the outflow rate toward the end of the membrane module header pipe compared with that at the inlet was attributable to the increase in the differential pressure between each membrane module toward the end of the header pipe.By applying the double-pipe theory to the existing header pipe (with an open-end inner pipe and a closed-end inner pipe), the evenness in the flow distribution improved. The CFD simulation and experimental results showed that evenness of the flow distributed improved by approximately 70% and 50%, respectively.When the double-pipe theory was applied to the inlet header pipe in the parallel-arrayed low-pressure membrane module, the evenness of the flow distribution improved. In future studies, the position of the reducer and the optimal diameter of the inner pipe orifices should be determined.

## Figures and Tables

**Figure 1 membranes-12-00720-f001:**
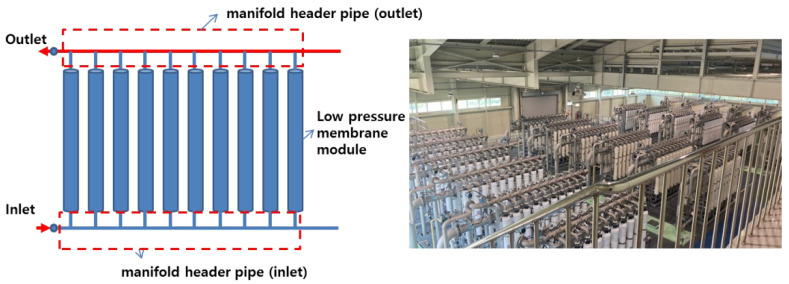
General parallel-arrayed low-pressure membrane module and pipes for water treatment and G drinking water treatment plant.

**Figure 2 membranes-12-00720-f002:**
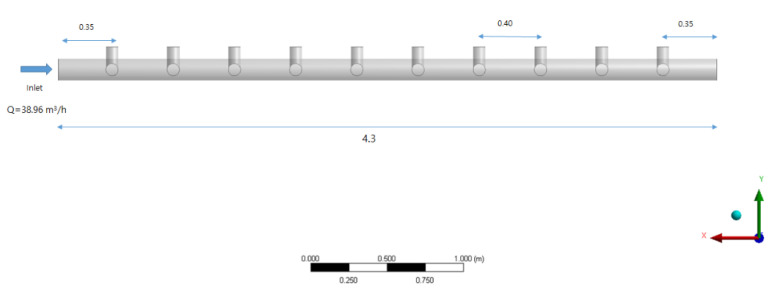
Geometry of inlet header and manifold pipes (units: m).

**Figure 3 membranes-12-00720-f003:**
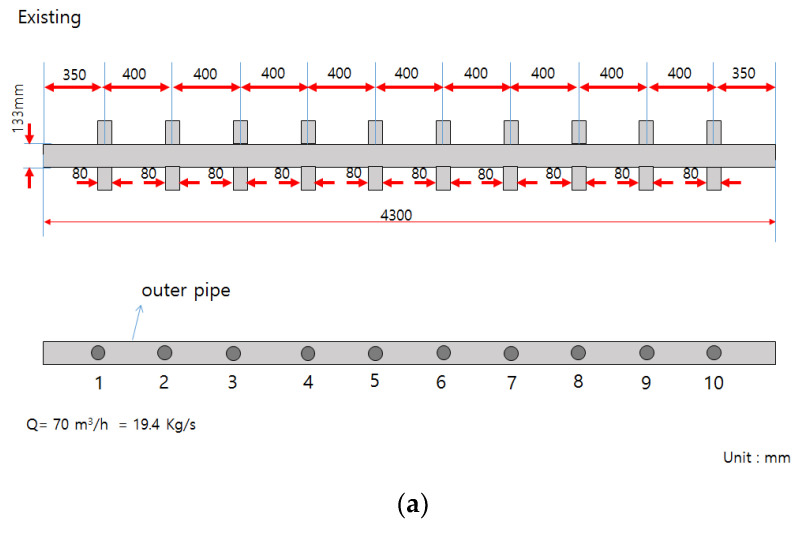
Geometry of inlet header manifold pipes: (**a**) existing pipe (single-manifold pipe); (**b**) double-manifold pipe (Case A, end open); (**c**) double-manifold pipe (Case B, end closed).

**Figure 4 membranes-12-00720-f004:**
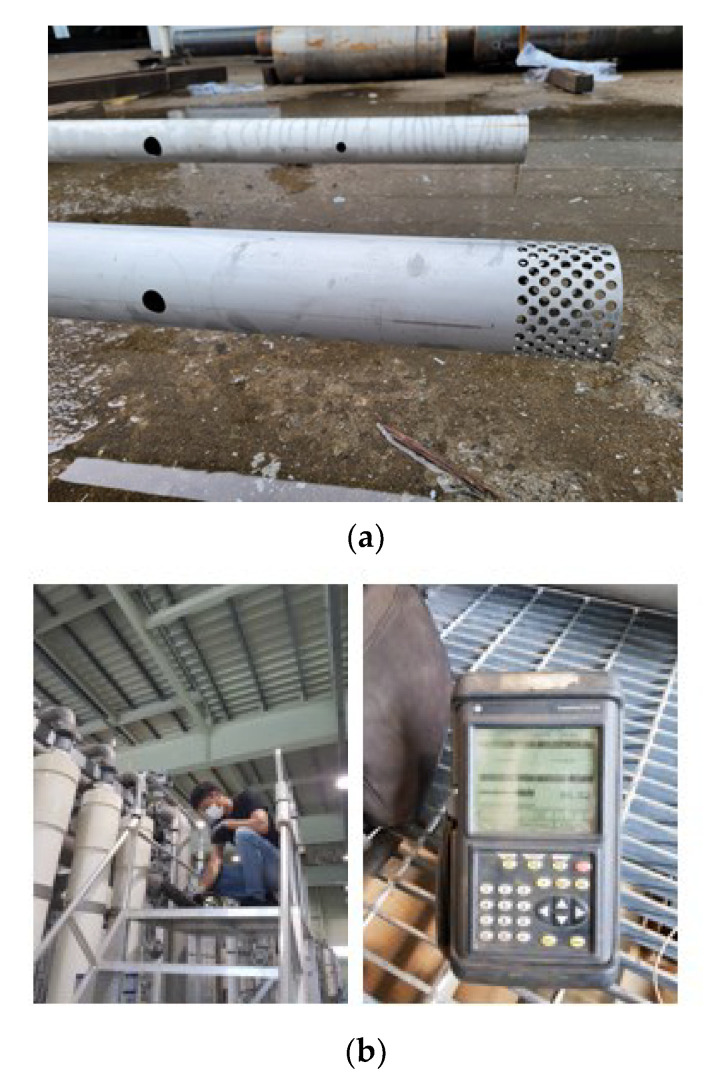
Inner orifice pipes and flow metering: (**a**) inner orifice pipes (Cases A and B); (**b**) metering using ultrasonic liquid flowmeter.

**Figure 5 membranes-12-00720-f005:**
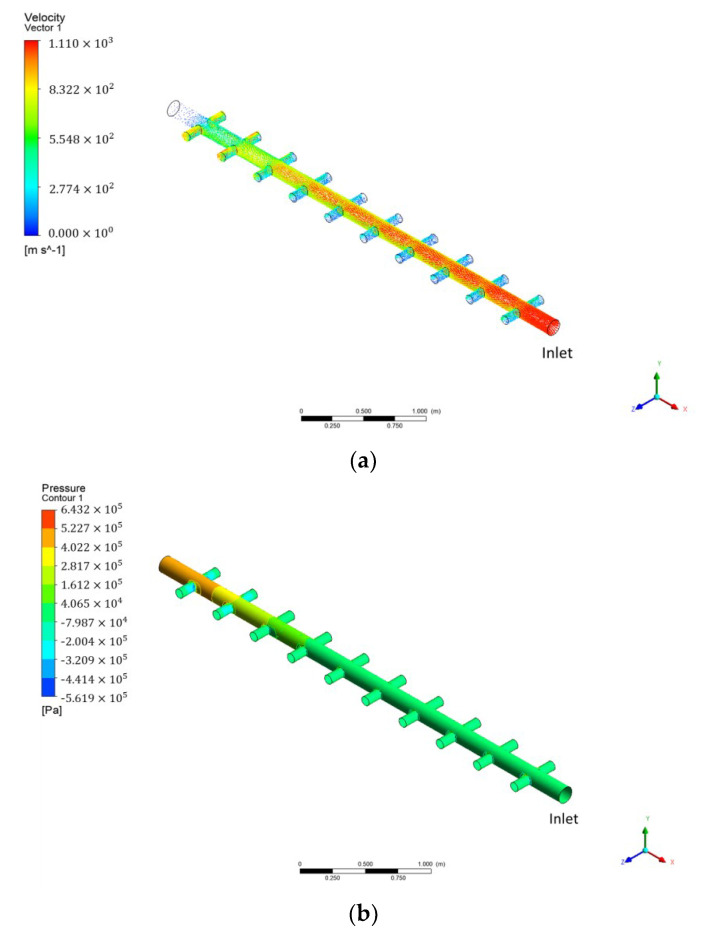
CFD simulation results for existing manifold pipe: (**a**) velocity vector; (**b**) pressure distribution contour.

**Figure 6 membranes-12-00720-f006:**
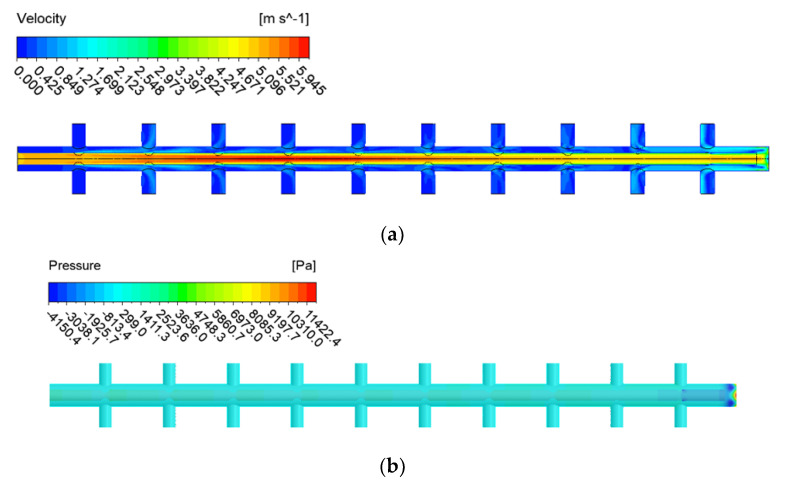
CFD simulation results for Case A (end open): (**a**) velocity contours; (**b**) pressure distribution contours.

**Figure 7 membranes-12-00720-f007:**
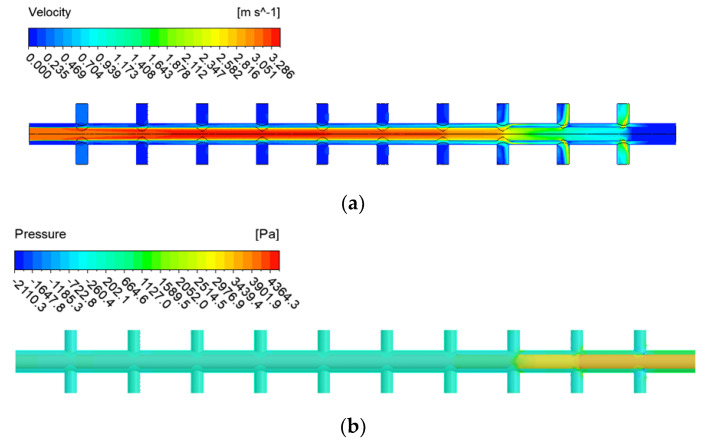
CFD simulation results for Case B (end closed with orifice wall): (**a**) velocity contour; (**b**) pressure distribution contour.

**Figure 8 membranes-12-00720-f008:**
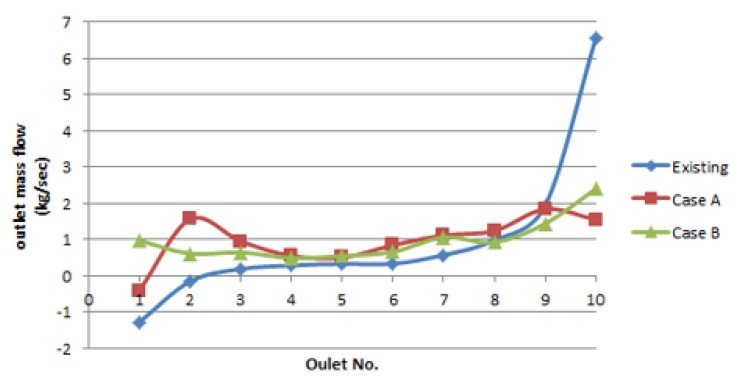
Comparison of CFD simulation results for all three cases.

**Figure 9 membranes-12-00720-f009:**
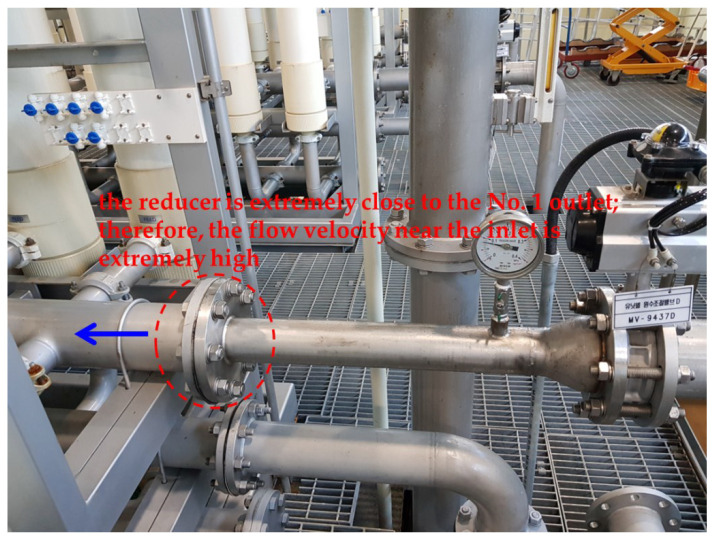
Location of reducer (the reducer is extremely close to the No. 1 outlet; therefore, the flow velocity near the inlet is extremely high).

**Table 1 membranes-12-00720-t001:** Membrane module specifications.

Membrane Manufacturer	Toray, HFS-2020
Membrane type	Microfiltration
Membrane module shape	External pressure-type hollow-fiber membrane (casing)
Hollow fiber	Inner D 0.9 mm/external D 1.4 mm
Pore size	0.05 µm
Membrane material	PVDF
Flux	(Ordinary) 1.0 m^3^/m^2^/day
(Max.) 1.33 m^3^/m^2^/day	
Module size	D 216 mm × L 2160 mm
Membrane area	72 m^2^/module
Allowable pressure	300 kPa
Allowable pH	1–10 during filtration, 1–12 during chemical cleaning

**Table 2 membranes-12-00720-t002:** Ultrasonic flowmeter and transducer specifications.

Flowmeter	PT878	Transducer Type	Clamp-On
Flow type	All acoustically conductive fluids	Applications	Liquid
Pipe size	12.7–7.6 m	Compatible meters	PT878
Pipe wall thickness	Maximum of 76.2 mm	Frequency	1 MHz
Pipe material	All metals and most plastics	Process temp.	−20 °C to 210 °C
Repeatability	±0.1% to 0.3% of reading	Ambient temp	−20 °C to 40 °C
Range	−12.2 to 12.2 m/s	Materials of construction	Metals and plastics
Range ability	400:1		
Measurement parameters	Volumetric flow, totalized flow, and flow velocity		

**Table 3 membranes-12-00720-t003:** Comparison of verification experiments results.

Outlet No.	Existing (m^3^/h)	Case A (m^3^/h)	Case B (m^3^/h)
1	2.28	2.44	2.46
2	2.49	3.1	3.13
3	2.5	3.31	3.28
4	2.66	3.1	2.97
5	2.59	2.91	3.2
6	3.55	3.1	3.0
7	3.22	2.57	2.49
8	3.21	2.85	2.39
9	3.34	3.31	3.0
10	3.9	2.98	3.2
Standard deviation	0.540	0.286	0.337

## Data Availability

Not applicable.

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
