# Peer review of "Application of Double Piping Theory to Parallel-Arrayed Low-Pressure Membrane Module Header Pipe and Experimental Verification of Flow Distribution Evenness"

_membranes, 2022, doi:10.3390/membranes12070720_

Round 1

Reviewer 1 Report

This paper apply double piping theory to improve flow distribution for parallel-arrayed low-pressure membrane. Several improvements are needed before this paper is accepted. A section after Introduction on double piping concept needs to be included. In Figure 9, why there are large error at number 2,3,4,6,7,8? These are not discussed in the paper. 

Author Response

Please, reference the attached file(Reply_for_Reviewer_1_Comments.docx).

Reviewer 2 Report

This manuscript can be recommended for publication after minor revision.

1) Fig. 1 has to be introduced in a higher quality. The text in Fig. 1 is not clear enough to be identified.

2) Thre references are not recent enough. No published references publised in 2022 are cited here. Many recent and relevent ones have to be cited in this manuscript.

Author Response

Please, reference the attached file(Reply_for_Reviewer_2_Comments.docx).

Author Response

Please, reference the attached file(Reply_for_Reviewer_3_Comments.docx).
